# Thoughtbubbles: an Unsupervised Method for Parallel Thinking in Latent Space

Houjun Liu[1]   Shikhar Murty[1][2]   Christopher D. Manning[1]   Róbert Csordás[1][3]

## Abstract

Current approaches for scaling inference-time compute in transformers train them to emit explicit chain-of-thought tokens before producing an answer. While these methods are powerful, they are limited because they cannot be applied during pretraining and rely solely on serially-generated, natural-language verbalization. In this work, we propose **Thoughtbubbles**, a transformer variant that natively performs parallel adaptive computation in latent space by learning to fork or delete residual streams. Thus, tokens requiring more computation can form a "bubble" of cloned residuals in the middle of the network. Crucially, this behavior is learned during pretraining with only language modeling loss. Using half of the training budget, **Thoughtbubbles** outperforms the perplexity and zero-shot evals of both standard decoder LMs and those using non-adaptive parallel computation approaches. These results hold across model sizes from 150M to 1.9B. **Thoughtbubbles** achieves competitive GSM8K results using half of the baseline's token budget. The implicit nature of our method enables models to begin learning adaptive computation at pretraining time, paving the way to unified train-time and test-time scaling behaviors.

## 1. Introduction

Despite their unprecedented success, Transformers (Vaswani et al., 2017) have a fixed computation budget and working memory, which present both a theoretical (Merrill & Sabharwal, 2023) and practical limit (Sanford et al., 2024) for solving complex, multi-step problems such as multi-hop retrieval or computer use agents.

[1]Department of Computer Science, Stanford University, Stanford, CA, United States, [2]DeepMind, New York, NY, work done while at Stanford [3]OpenAI, San Francisco, CA, work done while at Stanford. Correspondence to: Houjun Liu <houjun@stanford.edu>.

*Proceedings of the 43rd International Conference on Machine Learning*, Seoul, South Korea. PMLR 306, 2026. Copyright 2026 by the author(s).

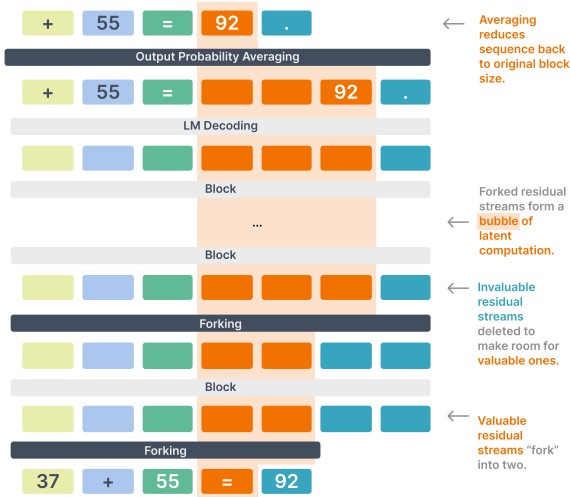

*Figure 1.* Overview of our method: input tokens fork to form a bubble of latent computation (orange), which is then contracted to produce the final token. Some extraneous tokens may fork (dark blue), but then be pruned.

Due to the growing interest in extending the capabilities of transformers for these difficult multi-step problems, many efforts are underway to surpass this bounded-computation limitation. The earliest and simplest is Chain of Thought (CoT) (Wei et al., 2023), which explicitly prompts the model to provide a set of reasoning steps. This technique allows the model to break a problem down into subproblems, solve them individually, and cache intermediate results for the full solution, thereby enabling a simple form of problem adaptivity (Merrill & Sabharwal, 2024).

Expanding upon this result, Pfau et al. (2024) shows both theoretically and practically that CoT improves transformers' expressiveness. This result holds when CoT traces are replaced with a unique thinking token at test time, indicating that simply adding residual streams, without explicit token-level reasoning, can improve computational performance.

Such an insertion of additional residual streams, the so-called "pause tokens," has since become a growing trend in recent architecture research. Though methods vary in terms of where to actually insert the thinking tokens (Herel & Mikolov, 2024; Sun et al., 2025; Goyal et al., 2024), all pause token approaches insert additional computation streams prior to inference. These streams are also applied to all layers, limiting the model's ability to allocate interme-

diate streams that are only useful in some layers (e.g., for computation that becomes useful after a few layers of attention have been applied). Furthermore, as Sun et al. (2025) notes, determining the location of pause tokens often requires manual design following the structure of the problem, which may be intractable for general language models.

In response, we present **Thoughtbubbles**, a novel Transformer-based architecture which enables the unsupervised and dynamic allocation of additional parallel residual streams for extra computation and memory. We achieve this by introducing a novel forking mechanism between some layers that computes and maintains a cumulative score for every residual stream and uses it to decide whether to *create* new residuals and *keep* existing ones.

This formulation makes dynamic computation a budget-bounded allocation problem of these scores. In order to train these scores to be useful, we use them to mask both the model's ability to attend to residual streams with low scores as well as limit the model's ability to update them at each layer. This attenuation forces the model to provide higher scores to residual streams it deems more important, which will also result in increased forking of those streams. At the end of encoding, our model will produce one output distribution for each stream by decoding each residual stream separately, including forked ones, and averaging the posterior probabilities weighted by their scores. We further find that a cheap approximation whereby the *residuals themselves* are averages still also confers similar expressivity advantages while giving a significant performance boost.

Thus, our approach will essentially create "bubbles" of latent computation consisting of forked residuals for difficult tokens (i.e., those with high cumulative scores) for additional thinking, before merging them to produce the final output token.

We conduct a variety of pretraining experiments across 150M to 1.9B scales and make the following contributions:

1. We introduce the first-known architecture to enable the unsupervised dynamic allocation of latent parallel computation, trainable as a regular decoder LM without any additional signal beyond language modeling loss.

2. We demonstrate that our approach performs better in all zero-shot evals as well as in GSM8K pass@1 using only *half of the training token budget* compared to baseline at 1.9B scale, and performance scales across 150M-1.9B scales

3. We further show that our method correctly allocates computation at *interpretable* regions of extra computation. Specifically, our method allocates more computation at regions of higher uncertainty (i.e., posterior entropy).

We release both PyTorch [1] and Jax [2] implementations for the community.

## 2. Methods

### 2.1. Overall Architecture

Our architecture is a decoder-only transformer (Radford et al., 2019), trained using the cross-entropy language modeling objective.

To achieve parallel computation, we want to allocate more residual streams corresponding to tokens that require more computation. To enable this, we propose a special type of operation named "forking", described in Section 2.3, which can duplicate or remove some residual streams for future computation.

The amount of forking is controlled by assigning a "cumulative score" between 0 and 1 to each residual stream. This score can be interpreted as the stream's existence indicator. In the forking operator, for each residual, the score is multiplied by two newly computed scores: "keep score" for updating the current stream, and a "fork score" indicating the importance of creating a new copy of this stream. We describe the computation of these scores in Section 2.3.

This setup reduces the dynamic computation task to determining which residual streams to keep or delete based on the value of cumulative scores: we take the top-k of the scores and keep their corresponding residuals (i.e., keep / fork). As long as the "useful" tokens receive the highest scores, the extra computation should help the performance of the model. To train the model to use the scores correctly, attention and residual updates are attenuated by the cumulative scores (Section 2.4). That is, the tokens that the model needs to attend to and update the most become implicitly the highest-scoring tokens to be duplicated.

Additionally, we take special care about the RoPE position embeddings: we apply a "partial rotation" to the forked tokens proportional to the number of forks: the more forks a token has, the "closer together" each of its forks are. This design is described in detail in Appendix F.

### 2.2. Notation

We will use $x_j^{(k)} \in \mathbb{R}^{d_{\text{model}}}$ to denote the $j^{\text{th}}$ residual stream at the $k^{\text{th}}$ layer. To emphasize that a particular token is the $j^{\text{th}}$ fork of token $i$, we will write $x_{i,j}^{(k)}$. We fork tokens to the left of the original input token. Thus, the original token is always $x_{i,0}^{(k)}$. A sequence of $q$ forks and original token can be written as $\left[ x_{i,q}^{(k)} \dots x_{i,0}^{(k)} \right]$.

---

[1] https://github.com/stanfordnlp/thoughtbubbles
[2] https://github.com/jemoka/fork-xla

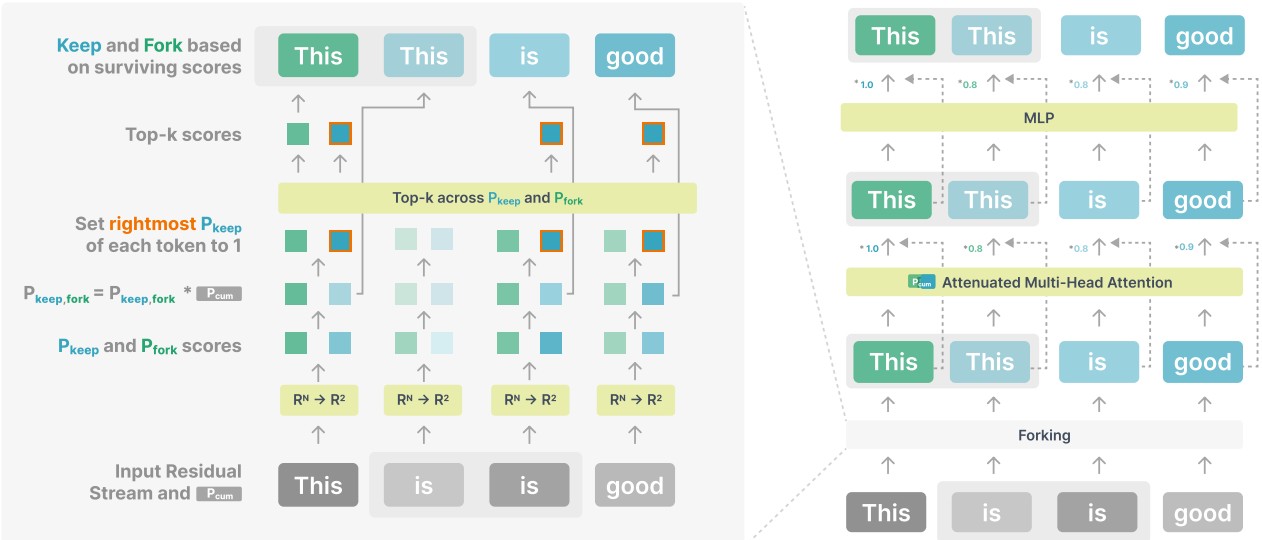

*Figure 2.* Forking procedure. Token "is" has two forks, one of which will get deleted; the token "this" creates a new fork; we show a score-attenuated transformer block after a forking operation.

Lastly, we use $L$ to denote the input sequence length (i.e., "input block size", the embedded input to the first block is $x_{1,0}^{(0)} \ldots x_{l,0}^{(0)}$), $N$ to denote the block size at the input to each layer (i.e., before the first layer, $N = L$). We omit the layer index for $N$ to avoid clutter. Additionally, we take a parameter $\kappa$ for the maximum block size. This means that the maximum number of forks at each layer is $\kappa - N$.

## 2.3. Forking

Residual stream insertion and deletion are performed in special forking layers inserted between our score-attenuated transformer blocks, described in Section 2.4. Each "forking" layer $k$ parametrized by $\theta$ carries a new "forking decision" function $f_\theta^{(k)} : \mathbb{R}^{d_{\text{model}}} \longrightarrow \mathbb{R}^2$. We apply this new function on each member of the residual stream in order to produce the fork and keep scores, which we then bottleneck using a top-k judgment in order to produce the forked output.

**Scoring.** For each residual, $x_i^{(k-1)}$ (note that the notation here is irrespective of forks or the original token, a distinction which we make later), we first apply the forking decision function along with a sigmoid activation $\sigma$ to obtain a fork and keep scores:

$$\sigma\left(f_\theta^{(k)}\left(x_i^{(k-1)}\right)\right) = \left[p_{\text{fork},i}^{(k)}, p_{\text{keep},i}^{(k)}\right]. \tag{1}$$

We then update the fork and keep scores inductively based on a "cumulative score" ($p_{\text{cum}}$) propagated from previous layers:

$$\hat{p}_{\text{fork},i}^{(k)} = p_{\text{cum},i}^{(k-1)} \cdot p_{\text{fork},i}^{(k)} \tag{2}$$

$$\hat{p}_{\text{keep},i}^{(k)} = p_{\text{cum},i}^{(k-1)} \cdot p_{\text{keep},i}^{(k)} \tag{3}$$

We initialize the cumulative scores for each input token at the first layer as $p_{\text{cum},(i,0)}^{(0)} = 1$. A subset of these $\hat{p}'_{\text{keep}}, \hat{p}_{\text{fork}}$ scores is used as $p_{\text{cum}}^{(k+1)}$, after deciding which ones to keep, as described later.

Note that, in practice, all scores (keep, fork, cumulative) are implemented in log-space for stability instead of being in probability space as shown here.

**Forking Judgments.** To make sure we have a source token from which to predict each next token, we must ensure that at least one instance is kept throughout the whole model. To do so, we first define a modified keep score that is forced to be 1 (the maximum) for the original, rightmost tokens:

$$\hat{p}'_{\text{keep},(k,j)} = \begin{cases} 1 \text{ if } j = 0 \\ \hat{p}_{\text{keep},(k,j)} \text{ otherwise} \end{cases} \tag{4}$$

Given a set of scores for a layer $k$, we create a list $P = \left[\hat{p}_{\text{fork},0}^{(k)}, \hat{p}_{\text{keep},0}^{'(k)} \cdots \hat{p}_{\text{fork},n}^{(k)}, \hat{p}_{\text{keep},n}^{'(k)}\right]$, we compute a top-k to shorten this list to obtain $P_\kappa$ where $|P_\kappa| = \kappa$. Using this list, we assemble the new residual stream set $X^{(k)}$ by the following two rules:

$$x_j^{(k)} \in X^{(k)} \text{ if } \hat{p}'_{\text{keep},j} \in P_\kappa \tag{5}$$

$$x_{j_{\text{fork}}}^{(k)} \in X^{(k)} \text{ if } \hat{p}_{\text{fork},j} \in P_\kappa \tag{6}$$

In order to differentiate the forks from their sources, a per-layer learned fork embedding $v_\theta'^{(k)} \in \mathbb{R}^{d_{\text{model}}}$ is added to their parent at initialization: $x_{j_{\text{fork}}}^{(k)} = x_j^{(k)} + v_\theta'^{(k)}$. We arrange the output tokens such that if a new forked residual is created, it is placed to the *left* of its parent.

We define the new cumulative scores $p_{\text{cum}}^{(k)}$ as $\hat{p}_{\text{fork},j}$ for newly forked residuals, and $\hat{p}_{\text{keep},j}$ for kept residuals (note that this is the score for which the rightmost token does not have a forced-maximum score of 1, allowing the model to ignore the rightmost token if desired.)

## 2.4. Residual Update Attenuation

To learn useful scores, in all blocks, both residual writes and attention computation are modulated by the cumulative scores. Intuitively, this prevents the model from relying on tokens that are about to be deleted due to their low scores.

Specifically, we stack the cumulative scores to a vector $P^{(k)} \in \mathbb{R}^\kappa$:

$$P^{(k)} = \left[ p_{\text{cum},1}^{(k)}, \dots, p_{\text{cum},\kappa}^{(k)} \right] \tag{7}$$

and use it to modulate both the attention computation and residual updates. We define the attenuated attention operation as:

$$\text{Attn}\left( Q^{(k)}, K^{(k)}, V^{(k)} \right) =$$
$$= \text{softmax}\left( \frac{Q^{(k)} K^{(k)\top} + \mathbf{1}\log\left( P^{(k)} \right)^\top}{\sqrt{d_{\text{model}}}} \right) \left( V^{(k)} \odot P^{(k)} \right) \tag{8}$$

where $\odot$ is the element-wise multiplication. We modify the transformer block (Vaswani et al., 2017) to attenuate the residual writes by $P^{(k)}$ as follows:

$$X^{(k)'} = \text{Attn}\left( f_Q(\text{LN}(X^{(k)})), f_K(\text{LN}(X^{(k)})), f_V(\text{LN}(X^{(k)})) \right) \\ \odot P^{(k)} \mathbf{1}^\top + X^{(k)} \tag{9}$$

$$X^{(k+1)} = \text{MLP}\left( \text{LN}\left( X^{(k)'} \right) \right) \odot P^{(k)} \mathbf{1}^\top + X^{(k)'} \tag{10}$$

for $X^{(k)}$ being the concatenated list of residual streams in the input of the layer, LN being layernorm, and $f_{Q,K,V}$ being the attention projections. If forking occurred prior to this layer, $X^{(k)}$ is as defined in eqs. 5 and Equation (6), *after* forking takes place.

## 2.5. Output Averaging

After all transformer layers, we obtain a residual stream set where an input token might be represented by multiple residual streams. To compute a single output distribution for these distributions, we decode each of the residual streams and mix the resulting probability distributions using the cumulative scores. For $\text{Dec}_\theta : \mathbb{R}_{\text{model}}^d \longrightarrow |V|$ being the

vocabulary output projection, and $f$ being the last layer of the network, we have:

$$x_i^{(k)} = \sum_j \frac{p_{\text{cum},(i,j)}^{(f)}}{\sum_l p_{\text{cum},(i,l)}^{(f)}} \text{softmax}\left( \text{Dec}_\theta \left( x_{i,j}^{(k)} \right) \right) \tag{11}$$

We compute this weighted average using the log-sum-exp trick (Blanchard et al., 2021) for stability.

Although the above formulation has a principled probabilistic interpretation and also avoids the softmax bottleneck (Yang et al., 2018), it is expensive both computationally and memory-wise due to the large vocabulary sizes typically used by LLMs. Thus, for our most expensive, 1.9B parameter model, we use the following cheap approximation instead:

$$x_i^{(k)} = \text{softmax}\left( \text{Dec}_\theta \left( \sum_j \frac{p_{\text{cum},(i,j)}^{(f)}}{\sum_l p_{\text{cum},(i,l)}^{(f)}} x_{i,j}^{(k)} \right) \right). \tag{12}$$

## 2.6. Scoring and Sampling

Because of the possibility of varying $\kappa$ at inference time, there are two main ways inference can be performed in our model. Naively, we can set the inference budget $\kappa_{\text{inference}}$ to be the same as in training time $\kappa_{\text{train}}$, two or four times the block size at training. We call this **fixed forking**. Alternatively, we can set the inference budget to be the same *ratio* as the training budget. $\kappa_{\text{inference}}$ is set to a value that maintains its same ratio to block size as during training; that is, if $\kappa_{\text{train}} = 2l_{\text{train}}$, then $\kappa_{\text{sample}} = 2l_{\text{sample}}$. We call this **dynamic forking**, and discuss this method further in Appendix H.1. Note that dynamic forking is essential for keeping the forking distribution close to the training while doing autoregressive generation. Thus, because of its impracticality, the fixed forking approach is not used for any evaluations in our results; all autoregressive outputs are measured via dynamic forking.

**Scoring** To obtain a probability judgment from our model of a sequence, we provide the entire sequence as input to our model and obtain the posterior probabilities our model assigns to each token of our sequence. For all of our results in Table 2, we use dynamic forking.

**Sampling** We perform autoregression with both fixed and dynamic forking, and discuss the tradeoffs of both, in Section 5.1. Note that dynamic forking is especially pertinent here because the initial sequence for autoregression is small.

## 3. Experimental Setup

### 3.1. Parameter Selection and Training

Because our architecture takes token embeddings as input and produces token probabilities, it trains exactly like a

regular decoder-only language model. As mentioned above, this means that the loss function can be standard language-modeling cross-entropy loss. Optimization is performed by the AdamW optimizer (Loshchilov & Hutter, 2017) with more details described in Appendix A.

We insert the first forking layer after a few regular transformer blocks to ensure that the forking score judgments see a broader context window. This is important in order to judge a token's relative importance compared to the others. For all models in Section 4, we train models at various scales with token forking placed prior to layers 3, 7, and 11. This means that for models with more layers, the majority of the latter half of the transformer will contain no forking. We discuss this choice in Appendix D.

### 3.2. Training Recipe

**1.9B Scale-Up Experiment** Our primary measurements are conducted using 1.9B parameter models trained by a standard two-stage training process. Our pre-training recipe, which we conduct for 40 billion tokens on a 10% warmup-stable schedule followed by a constant learning rate. The training data is a mixture of 70% FineWeb (Penedo et al., 2024), and 30% peS2o (Soldaini & Lo, 2023) to create an English-dominant high-quality scientific text corpora. We then cool down these checkpoints using a cosine decay to 10% of the original learning rate using a mid-training recipe, which involves a 2 billion token mixture of 5% MMLU (Hendrycks et al., 2021), 30% SmolTalk (Allal et al., 2025), 25% of the pretraining mix, and finally 40% GSM8K-aug—a synthetic math reasoning corpus widely used in small-scale reasoning literature (Ali et al., 2024; Shen et al., 2025; Kong et al., 2025). This forms a standard warmup-stable-decay schedule (Wen et al., 2025) used in current pretraining paradigms.

We checkpoint our model both at the end of pretraining as well as halfway through training, at 20B tokens—allowing the fully-trained baseline to be a generous *over-estimate* of both computation and data compared to our approach. The constant learning rate enables us to generate both of these checkpoints from the same training run.

**150M-772M Scaling Suite** To demonstrate the scaling behavior of our approach, we pretrain our approach ranging from 150M - 772M scales on two datasets: OpenWebText (Gokaslan et al., 2019), a standard web-text pretraining corpus, as well as peS2o (Soldaini & Lo, 2023), a collection of academic papers sourced from the Semantic Scholar Open Research corpus (Lo et al., 2020). Pretraining is conducted for 2.5 billion tokens. These models use a more traditional cosine decay learning rate schedule.

### 3.3. Baselines

**Regular Transformer.** We first compare against a GPT-2-like (Radford et al., 2019) transformer with RoPE (Su et al., 2024). Our model is based on nanoGPT[3]. We make no changes other than removing the learned position embeddings and including rotational ones in the attention pass.

**Duplicated Filler Tokens.** Though a regular transformer is a parameter-matched baseline, our approach will necessarily utilize more computation due to the expanded latent block size (i.e., after forking, the block-size is longer). A naive model of parallel computation that would allow us to slightly exceed the computation of our approach is by simply copying the input residual multiple times before running the transformer, and then taking the rightmost residual for decoding.

### 3.4. Pretraining Evaluations

After pretraining, we conduct a variety of zero-shot evaluations on our models and baselines to examine their quality. They include the model's measured perplexity on a holdout validation set, LAMBADA (Paperno et al., 2016) for context extraction, HellaSwag (Zellers et al., 2019) for common sense reasoning, BLiMP (Warstadt et al., 2020) for syntax understanding, AI2-ARC (Clark et al., 2018) for basic reasoning, and PIQA (Bisk et al., 2020) for embodied physical inference. For each zero-shot downstream task, we use the dynamic budget as described in Appendix H.1. We describe in detail the implementation of the zero-shot evaluations, including what to measure, in Appendix H. Across all evaluations, we use the trained models as-is without additional fine-tuning. For our full-scale model, after mid-training, we additionally evaluate our model and baselines' zero-shot chain-of-thought performance on GSM8K's test set (Cobbe et al., 2021).

## 4. Results

**Even at half the token budget, at 1.9B scale, our model achieves better performance against all evaluation metrics.** Our model, as highlighted in Table 1, achieves better performance against baseline in all zero-shot evaluation metrics *as well* as on the reasoning dataset GSM8K. Excitingly, we reached or exceeded baseline performance by only using half of the token budget (i.e., half of the pretraining time), giving a generous lower-bound on our performance.

**Our approach performs the best against all baselines in validation perplexity, even exceeding models of bigger scale.** Across both parameter and computation matched settings, we find that our model scores the lowest perplexity

---

[3] https://github.com/karpathy/nanoGPT

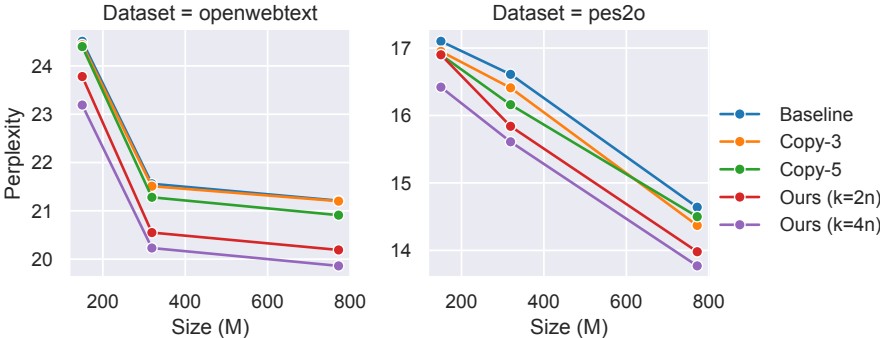

*Figure 3.* Dev-set perplexity of our approach and various baselines as a function of model scale on both `OpenWebText` and `peS2o` datasets. Across all scales, our method outperforms all baselines, including both computation and parameter-matched ones. Lower is better.

| | Ours ($\kappa = 2L$) | | Baseline |
|---|---|---|---|
| | 20B tokens | 40B tokens | 40B tokens |
| Perplexity ($\downarrow$) | 12.68 | 12.09 | 15.03 |
| BLiMP ($\uparrow$) | **78.54** | 78.41 | 77.49 |
| HellaSwag ($\uparrow$) | 50.04 | **54.16** | 47.29 |
| PIQA ($\uparrow$) | **73.42** | 73.00 | 71.79 |
| ARC-Easy ($\uparrow$) | 39.72 | **42.53** | 38.31 |
| ARC-Challenge ($\uparrow$) | 24.50 | **29.53** | 26.17 |
| LAMBADA ($\uparrow$) | 39.32 | **43.91** | 39.15 |
| GSM8K ($\uparrow$) | 31.50 | **32.10** | 31.46 |

*Table 1.* Zero-shot evaluation results across all model scales after pretraining on 20 and 40 billion tokens. Each setting is parameter-matched at 1.9B parameters. Baseline is a standard GPT-2-like model; ours is the **thoughtbubbles** transformer, with forking budget set to 2x ($\kappa = 2L$) the input block size. Measuring 20B tokens of pretraining of our approach against 40B tokens of baseline gives a generous *temporal*, *data* and *computation* lower bound.

across all evaluations. Figure 3 highlights the scalability of our approach: surprisingly, our approach at a 319M parameter scale has lower perplexity on `OpenWebText` than the baseline approach at the 772M scale.

**Our approach's performance dominates at scale.** Our approach's performance dominates all baselines, and this effect becomes more prominent as model size scales. Across all scales, we find that our approach confers a performance gain in all LAMBADA and HellaSwag evaluations in both the parameter-matched baselines as well as the computation-matched baselines. However, we note that for BLiMP (syntax understanding) our model only outperforms the parameter-matched, but not computation-matched baselines—indicating that dynamic parallel computation may not be as helpful for syntax matches.

## 5. Analysis

Here we describe a series of analyses of our approach at the 772M scale to understand its capabilities and behaviors. We further describe ablations to each component of our approach in Appendix C, in particular to understand the token merging behavior (Table 6) and our attention masking approach (Table 7). Finally, we measure and describe our wall-clock performance in Appendix B, noting that our performance gains are also competitive to timing-based measurements.

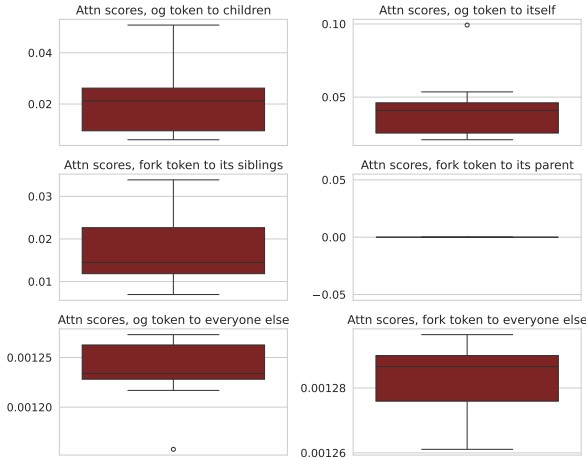

*Figure 4.* Analysis of attention allocation between the main (rightmost, "og") token and its child forks on our approach trained on `openwebtext`. Note that since we place child token embeddings to the to the left of the main token, forked children cannot attend to its parent.

**Forks meaningfully influence the value of the parent token.** In Figure 4, we see the rightmost ("og") token attends to its children with attention scores more than an order of magnitude higher than other tokens—second only to attention of those tokens to themselves. This result indicates that the forking tokens play a larger role in the computation of the residual update for the rightmost token than most other

| Dataset | Size | Approach | Perplexity (↓) | LAMBADA (↑) | HellaSwag (↑) | BLiMP (↑) | PIQA (↑) |
|---|---|---|---|---|---|---|---|
| OpenWebText | 772M | Baseline | 21.22 | 23.9 | 30.6 | 79.6 | **62.3** |
| | | Copy-3 | 21.20 | 22.8 | 29.0 | 81.2 | 60.4 |
| | | Copy-5 | 20.90 | 19.9 | 29.1 | 80.9 | 60.2 |
| | | Ours ($\kappa = 2L$) | 20.19 | 27.9 | 31.1 | 80.4 | 62.0 |
| | | Ours ($\kappa = 4L$) | **19.74** | **29.4** | **32.25** | **81.6** | 61.9 |
| | 319M | Baseline | 21.56 | 22.1 | 28.7 | 79.0 | 60.5 |
| | | Copy-3 | 21.51 | 21.9 | 28.6 | **80.5** | 60.1 |
| | | Copy-5 | 21.28 | 21.1 | 28.4 | 79.6 | 60.5 |
| | | Ours ($\kappa = 2L$) | 20.55 | 22.9 | **29.3** | 78.3 | **60.9** |
| | | Ours ($\kappa = 4L$) | **20.23** | **23.2** | 29.0 | 78.8 | 60.1 |
| | 150M | Baseline | 24.51 | 18.2 | 26.9 | 76.7 | 57.9 |
| | | Copy-3 | 24.44 | 17.6 | 27.1 | **79.3** | 58.9 |
| | | Copy-5 | 24.40 | 18.9 | 26.9 | 78.8 | 59.4 |
| | | Ours ($\kappa = 2L$) | 23.78 | 21.1 | 27.3 | 77.5 | 59.0 |
| | | Ours ($\kappa = 4L$) | **23.19** | **25.5** | **27.7** | 78.1 | **60.6** |
| peS2o | 772M | Baseline | 14.64 | 9.9 | 27.3 | 69.8 | 55.4 |
| | | Copy-3 | 14.37 | 9.5 | 27.2 | **73.3** | 55.3 |
| | | Copy-5 | 14.50 | 10.3 | 27.3 | 71.6 | 54.5 |
| | | Ours ($\kappa = 2L$) | 13.98 | 10.5 | 27.4 | 68.4 | **56.3** |
| | | Ours ($\kappa = 4L$) | **13.77** | **12.9** | **27.6** | 67.4 | 54.6 |
| | 319M | Baseline | 16.61 | 9.3 | 26.4 | 68.4 | **55.3** |
| | | Copy-3 | 16.41 | 9.4 | **27.2** | **71.8** | 54.7 |
| | | Copy-5 | 16.16 | 8.5 | 26.6 | 70.1 | 55.1 |
| | | Ours ($\kappa = 2L$) | 15.84 | 10.5 | 26.5 | 67.0 | 53.8 |
| | | Ours ($\kappa = 4L$) | **15.61** | **12.3** | **27.2** | 68.6 | 53.6 |
| | 150M | Baseline | 17.10 | 8.1 | 26.4 | 68.6 | 54.5 |
| | | Copy-3 | 16.95 | 7.1 | 26.3 | **69.6** | 54.1 |
| | | Copy-5 | 16.90 | 7.2 | 26.0 | 69.3 | 54.0 |
| | | Ours ($\kappa = 2L$) | 16.90 | 5.0 | 26.2 | 66.6 | **55.1** |
| | | Ours ($\kappa = 4L$) | **16.42** | **10.3** | **26.9** | 67.9 | **55.1** |

*Table 2.* Zero-shot evaluation results across all model scales after pretraining on 2.5 billion tokens. Each setting is parameter-matched; baseline is a standard GPT-2-like model; copy-3 and copy-5 are models where the input residuals are copied multiple times and can attend to each other; ours is the **thoughtbubbles** transformer, with forking budget set to 2x ($\kappa = 2L$) and 4x ($\kappa = 4L$) the input block size.

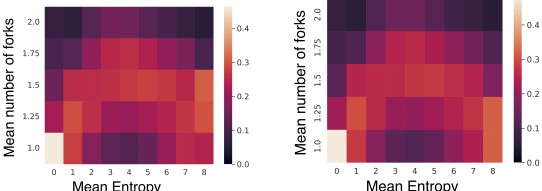

*Figure 5.* Normalized number of forks in the final layer across a window of 4 tokens as a function of the mean entropy of those 4 tokens on `OpenWebText`. Left: entropy as measured by the forking transformer; right: entropy as measured by a baseline decoder LM. Note that our model independently induced the entropy ratings of an unrelated decoder LM.

tokens, indicating their utility in computing the final output.

**Our model allocates more computation at regions of higher uncertainty without explicit supervision ...** Despite no explicit interventions or regularization, our method learned to allocate more computation at areas of greater uncertainty. We see in Figure 5 that our method allocates more tokens with high output entropy; this is true both for the entropy measured from the forking model as well as an independently trained, parameter-matched decoder LM that does not fork. This is in line with recent literature (Wang et al., 2025) that highlights the informativeness of high entropy tokens.

**... but will reduce computation at areas of greatest uncertainty.** Despite the previous point, however, we note that our model allocates relatively less budget at tokens of the highest uncertainty, forming a concave parabolic relationship between entropy and computation allocation. We hypothesize that this is due to the relatively higher utility of further computation at areas of moderate (but not low) uncertainty: for instance, while choosing between a few options; conversely, areas of highest uncertainty are often caused by the edges of clauses or coreferences, where additional computation will not help resolve the uncertainty.

### 5.1. Autoregression

As seen in Figure 6, implementing autoregression naively with a fixed block size irrespective of the input sequence length results in a distribution shift between blockwise forward pass and autoregression—since the maximum allowed number of forks is much higher if input sequence length is smaller while the total budget remains the same.

However, if we apply the forking budget scaling mitigation described in Appendix H.1, we find that our model performs roughly equivalently to the blockwise forward pass, and retains our approach's performance gains above baseline. This result indicates that, while our result can adapt to different

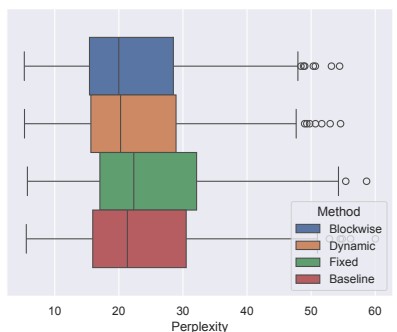

| Approach | Perplexity |
|---|---|
| Ours (Blockwise) | **20.97** |
| Ours (Fixed Budget) | 23.10 |
| Ours (Dynamic Budget) | 21.18 |
| Baseline | 22.15 |

*Figure 6.* Perplexity distribution and mean perplexity of our 772M ($\kappa = 2L$) model over smaller subset of `OpenWebText` dev set between blockwise forward versus autoregression. Left: naive autoregression; right: autoregression with forking budget proportional to input size. Lower is better.

inference-time input sizes, care must be taken to scale the adaptive computation budget accordingly.

## 6. Related Work

We provide here a contextualization of our approach and a few predominant families of approaches. We include additional discussion of specific peer approaches in Appendix I.

**Chain-of-Thought Approaches**   Chain-of-thought (Wei et al., 2023) is a simple form of adaptive computation which uses natural-language-based autoregression with additional tokens to achieve thinking. Variants of this approach include simply supervising the output chain (Zhang & Ding, 2024), to replacing them with continuous traces (Hao et al., 2024; Zelikman et al., 2024) or controlled non-adaptive filler tokens (Pfau et al., 2024). Unlike chain-of-thought, our method performs adaptive computation not with recurrence but parallel computation, improving efficiency as well as being able to be trained without additional supervision.

**Adaptive Computation**   Methods vary to force a dynamic amount of computation from a neural model based on the problem. The oldest approaches involve explicitly forcing recurrence (Graves, 2016), while modern LMs yield performance improvements through forcing very simple interventions to existing chain of thoughts (Muennighoff et al., 2025), skipping or adding recurrent compute across layers without adding additional streams of computation (Dehghani et al., 2019; Murty et al., 2023; Csordás et al., 2024; Chen et al., 2024; Raposo et al., 2024; Kallini et al.,

2024), or by adding additional residual streams when computation is needed (Herel & Mikolov, 2024; Goyal et al., 2024; Sun et al., 2025). Our method removes the need to insert latent tokens explicitly during training or inference, but still gives the ability to gain additional streams of computation through latent residual streams.

**Analysis of Latent Computation**   There's a large and robust literature on the complexity-theoretic power of transformers. Results have shown the limited expressive power of standard transformer computation (Merrill & Sabharwal, 2023), and the additional power that chain-of-thought or even padding tokens add to the computation (Merrill & Sabharwal, 2025; London & Kanade, 2025). Work has also shown the limitations given by single special-token thinking approaches that are not input adaptive (Vennam et al., 2024). Prior work has also shown through techniques in interpretability that even simple chain of thought computation carries implicit intermediate computation similar to depth-bounded recurrence (Brinkmann et al., 2024). We also demonstrate here the power of adaptive latent computation in our work by demonstrating its superior performance even against computation matched baselines; furthermore, we demonstrate that we are indeed performing additional computation in "decisive" high entropy tokens, in line with prior analyses (Wang et al., 2025).

## 7. Conclusion

In this work we introduce **thoughtbubbles**, the first adaptive parallel computation architecture that's 1) trainable without additional supervision beyond LM loss 2) allocates computation and memory at interpretable regions of uncertainty and 3) performs better than baseline models in both perplexity and across a suite of zero-shot evals on both parameter-matched and computation-matched settings.

This method unlocks a form of input-adaptivity in transformer computation, which allows our model to solve more difficult tasks that require scaling inference-time computation. We demonstrate the efficacy of our method via a suite of zero-shot evaluations as well as gsm8k evaluations in both computation and parameter matched settings at 1.9B parameters; furthermore, in our scaling experiments between 150M-772M parameters, we discovered that our method at a smaller 319M scale outperformed baselines at 772M scale.

Most importantly, our model enables learning latent adaptive computation in a language model during the pre-training phase. Our approach does not rely on being exposed to step-wise instructions during pre-training. We hope that this will unlock a new generation of transformer architectures with more general latent computation—a step towards enhancing input-adaptive methods like chain-of-thoughts to be fully latent and integrated as a part of pretraining.

## Impact Statement

This paper presents work aimed at advancing the state of the art in neural network architectures. There are many potential societal consequences of our work that come with any general improvements to neural network capability, none of which we believe must be specifically highlighted here.

## Acknowledgements

We thank Mykel Kochenderfer, Christopher Potts, Tatsunori Hashimoto, Percy Liang, Diyi Yang, Jacob Andreas, and David Bau for their insightful comments and discussion of this project. In addition, we want to thank our friends and colleagues at Stanford NLP and the Swiss AI Lab ID-SIA for our many interesting discussions regarding this work, especially Julie Kallini, Dilara Soylu, Steven Cao, Amelia Hardy, Liam Kruse, Sydney Katz, Vincent Herrmann, Zachary Sayyah, Huxley Marvit, Joseph Shetaye, and Eric Alcaide.

Christopher D. Manning is a CIFAR Fellow. This research was supported with Cloud TPUs from Google's TPU Research Cloud (TRC).

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

# A. Exact Benchmark Model Architecture

Our benchmark runs involve a variety of model configurations across different scales. All models were trained with a shared optimization configuration, detailed in Table 3. Optimization was performed using mix-precision training using `bfloat16`, but with the cumulative forking scores tracked in log space in `float32`; we chose to do this in particular because small numerical imprecision forking judgments may result in large top-k outcome differences.

A vocab size of 50304 to optimize for tensor core efficiency is used, resulting in 47 unused tokens. Forking layers are placed in layers 3, 7, and 11–irrespective of $N_{\text{layers}}$ of the design.

| Hyperparameter | Value |
|---|---|
| Maximum Learning rate | 2.5e-4 |
| Warmup fraction | 0.01 |
| Optimizer | AdamW |
| Weight decay | 0.1 |
| Warmup | 0.01 |
| $\beta_1$ | 0.9 |
| $\beta_2$ | 0.95 |
| Dropout | 0.0 |
| Bias | True |
| Batch size (global) | 64 (150M-772M); 480 (1.9B) |
| Tokens | 2.5B (150-772M); 40B (1.9B) |
| Vocab size | 50304 |
| Block size | 512 |

*Table 3.* Optimization parameters shared across all scales; note that the actual per-machine batch size differs based on model architecture, the details of which is listed below.

Each model scale shares a common implementation, but contains different topology configurations which increase its parameter count; these configurations are enumerated in Table 4.

| Size | Approach | $N_{\text{layers}}$ | $N_{\text{heads}}$ | $d_{\text{model}}$ | Batch | Accumulation | Expanded Size |
|---|---|---|---|---|---|---|---|
| 150M | Baseline | 16 | 12 | 768 | 8 | 8 | 512 |
| 150M | Copy-3 | 16 | 12 | 768 | 8 | 8 | 1536 |
| 150M | Copy-5 | 16 | 12 | 768 | 8 | 8 | 2560 |
| 150M | Ours ($\kappa = 2L$) | 16 | 12 | 768 | 8 | 8 | 1024 |
| 150M | Ours ($\kappa = 4L$) | 16 | 12 | 768 | 8 | 8 | 2048 |
| 319M | Baseline | 24 | 16 | 1024 | 4 | 16 | 512 |
| 319M | Copy-3 | 24 | 16 | 1024 | 4 | 16 | 1536 |
| 319M | Copy-5 | 24 | 16 | 1024 | 4 | 16 | 2560 |
| 319M | Ours ($\kappa = 2L$) | 24 | 16 | 1024 | 4 | 16 | 1024 |
| 319M | Ours ($\kappa = 4L$) | 24 | 16 | 1024 | 4 | 16 | 2048 |
| 772M | Baseline | 36 | 20 | 1280 | 2 | 32 | 512 |
| 772M | Copy-3 | 36 | 20 | 1280 | 2 | 32 | 1536 |
| 772M | Copy-5 | 36 | 20 | 1280 | 2 | 32 | 2560 |
| 772M | Ours ($\kappa = 2L$) | 36 | 20 | 1280 | 2 | 32 | 1024 |
| 772M | Ours ($\kappa = 4L$) | 36 | 20 | 1280 | 2 | 32 | 2048 |
| 1.9B | Baseline | 36 | 16 | 2048 | 14 | 32 | 512 |
| 1.9B | Ours ($\kappa = 2L$) | 36 | 16 | 2048 | 14 | 32 | 1024 |

*Table 4.* Model topology parameters for each scale

Optimization of each run at 150M-772M scale is conducted on a single NVIDIA H200 GPU. Optimization of each run at 1.9B scale is conducted on a single v4-32 TPU pod. Dataset tokenization uses the pre-trained tokenizer from GPT-2 (Radford et al., 2019). FlashAttention kernels (Dao et al., 2022) and XLA attention kernels, where appropriate, are used to train our system, with value vector attenuation occurring before.

# B. Training and Inference Wall-Clock Efficiency

We find that, with an efficient implementation of our approach, and in particular the optimization of residual averaging described in Section 2.5, our forking mechanism is extremely lightweight in terms of wall-clock efficiency—almost matching those of a baseline transformer.

In particular, using `torch.compile` graph lowering and sequentially blocking CUDA operations (to gain the most accurate timing signals), we obtain the following wall clock performance speeds for a single forward-backward pass of 8 sequences:

| Method | Time (ms) |
|---|---|
| Baseline | 234 |
| Ours ($\kappa = 2L$) | 327 |
| Baseline (2L) | 380 |

*Table 5.* Wall-clock performance on a single Nvidia H100, with blocking CUDA kernel launch operations, for our approach at $1.9B$ scale and on a batch of 8 sequences; $L = 512$ is the block size used.

We find that our full forward/backward pass chain almost matches the efficiency of a baseline transformer, and in particular is *faster* than a model trained with double the block size. These results, combined with the fact that our approach outperforms baseline at only *half* of the token count (Table 1), implies competitive wall-clock performance advantage of our approach as well.

This empirical wall-clock efficiency is also confirmed by our theoretical analysis in Appendix G, which demonstrates that our approach with double the forking indeed only incurs around 2 times the FLOP/s.

# C. Gradient Signal Ablations

**Use of Forked Tokens' Gradient Signals** We perform a minimal ablation on our 150M model to analyze whether our model is actually applying gradients to all forking tokens (i.e., instead of ignoring them); in Table 6, we perform a minimal ablation where we keep only one, namely the rightmost, residual channel per input token instead of averaging all forked tokens. We find that this performs worse than our approach.

| Approach | Number of Tokens (OpenWebText) | Validation Loss |
|---|---|---|
| Baseline | 4.8B | 3.27 |
| Ours ($\kappa = 2L$, logit avg.) | 2.4B | 3.17 |
| Ours ($\kappa = 2L$, keep rightmost) | 2.4B | 3.21 |

*Table 6.* Performance of an ablation where we keep the rightmost token instead of logit average for output token merging.

**Use of Attention Masking Gradient Signals** We additionally perform an ablation to analyze whether our model is using the gradient signal from masking the $QK$ as well as $V$ in attention computation.

| Approach | Number of Tokens (OpenWebText) | Validation Loss |
|---|---|---|
| Baseline | 4.8B | 3.27 |
| Ours ($\kappa = 2L$, logit avg.) | 2.4B | 3.17 |
| Ours ($\kappa = 2L$, don't mask attn) | 2.4B | 3.90 |

*Table 7.* Ablation: removing attention masking significantly degrades performance.

We believe this degradation is due to the fact that cumulative scores are multiplicative; without them affecting computation in some way, the multiplicative scores across layers would simply decrease throughout computation. Thus, the top-$k$ decision is essentially random, resulting in random forks and thus significantly decreased performance.

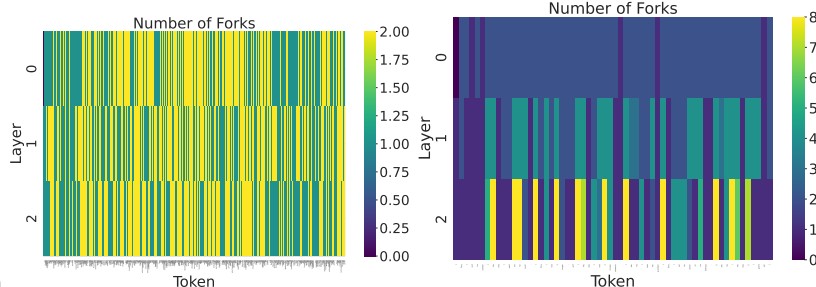

*Figure 7.* Number of token forks created by the model at each layer for an input sample of `OpenWebText` (top), and number of forking tokens created by the model on a sample of `CLUTTR` (bottom).

## D. Overforking

We perform a minimal ablation to examine if additional layers of forking beyond layers 3, 7, and 11 would help model performance. In particular, we trained our 772M scale model for 25,000 steps with forking at layers 3, 7, and 11 only as well as extended into all layers $4n - 1$ (i.e., 16, 20, ...) thereafter. We find that forking more confers only a slight advantage to performance; we believe this is due to certain tokens with high cumulative scores early on in the model being dropped by hard top-k decisions later in the model, thus resulting in no gradients to update the early large cumulative scores. By implementing training time randomization and noise, this can be mitigated to improve deep forking performance.

| Approach | Perplexity |
|---|---|
| Ours | 29.84 |
| Ours (extended forking) | 28.02 |

*Table 8.* Performance of an ablation where we performed more forking at later layers, trained on `OpenWebText` for 25,000 steps (roughly 0.8BT). We see that the extended forking approach is only slightly better than forking only in the beginning.

## E. Analysis of Forking Locations

We perform here a qualitative analysis of where our model allocates computation. In particular, after training, we plot the number of residuals each input token is forked into after each forking judgment. We run this analysis on both a sample of `OpenWebText` , as well as a synthetic task with known "difficult" computation locations—a relational graph ST-connectivity task named `CLUTTR` (Sinha et al., 2019).

In Figure 7, we qualitatively observe that our model has learned to perform extra computation near interpretable decision boundaries for synthetic tasks. For `CLUTTR`, forking occurs near coreferent entities and at special tokens delineating the beginning of query components or response. In contrast, the result for `OpenWebText` shows that computation on web-text is spread evenly across the sequence—namely, that it's not sequence position dependent, as it does for synthetic tasks such as `CLUTTR` due to their structure.

This result, along with the result in Figure 5, indicates that our model can truly dynamically allocate computation to the areas of the greatest computational difficulty and is not relying on a simple heuristic for allocating computation.

## F. Position Encoding

Due to the fact that this architecture introduces multiple possible residuals for every input token, care must be taken to ensure that position embeddings scale by the amount of forking accordingly. In order to do this, we implement a Rotational Position Embedding (RoPE, Su et al. (2024)) variant to offset smaller rotations degrees when there are more forks.

Recall that, typically, RoPE is defined, for $x_k^{(j)}$ being the $j$-th slot of the residual stream of token $k$,

$$\text{RoPE}\left(x_k^{(i)}, x_k^{(j)}, k\right) = \begin{pmatrix} \cos k\theta & -\sin k\theta \\ \sin k\theta & \cos k\theta \end{pmatrix} \begin{pmatrix} x_k^{(i)} \\ x_k^{(j)} \end{pmatrix}. \tag{13}$$

where $\theta$ is the total rotation angle. In our approach, however, we may have $q$ streams representing a particular input token. That is, token $k$ is forked into residual streams $x_{(q-1),k}, \ldots, x_{0,k}$. In order to accommodate tokens of different number of forks, we augment RoPE with *partial* rotations proportional to the number of forks of each token. For the $i, j$-th slot of residual $p$ of token $k$ which has $q$ forks in total, we write:

$$\text{RoPE}\left(x_{p,k}^{(i)}, x_{p,k}^{(j)}, k\right) = \begin{pmatrix} \cos\left((k - \frac{p}{q})\theta\right) & -\sin\left((k - \frac{p}{q})\theta\right) \\ \sin\left((k - \frac{p}{q})\theta\right) & \cos\left((k - \frac{p}{q})\theta\right) \end{pmatrix} \begin{pmatrix} x_k^{(i)} \\ x_k^{(j)} \end{pmatrix}. \tag{14}$$

That is, the more forks a particular token has, the "closer together" in position embeddings each of its forks will be.

## G. Detailed FLOP/s Analysis

Let $d$ be our hidden dimension and $L$ be the block size: attention projections cost $6Ld^2$, output projection costs $2Ld^2$, attention itself is quadratic in $4L^2d$, MLP for $16Ld^2$.

The only mechanism is a forking projection, which maps inputs $L \times d$ to a $d \times 2$ map; with the convention that multiplication counts as one pass and addition counts as another, we obtain that the forking projection costs $4Ld$.

Let the forking model's block size be $kL$. Thus the training performance ratio (baseline FLOPs / our FLOPs each block) would be:

$$\frac{24Ld^2 + 4L^2d}{24kLd^2 + 4k^2L^2d + 4kLd} = \frac{6d + L}{k\left(6d + kL + 1\right)} \tag{15}$$

setting $d = 2048$, $L = 512$, $k = 2$, this would be around $0.481$, which as expected is around 2 times the FLOPs. In terms of inference time FLOPs, with an assumption of caching, each token would attend to two times the cached key-value pairs, thus an analysis would follow similarly.

## H. Details on Zero-Shot Evals

**Perplexity**   We first evaluate the perplexity score of each model against the development sets of the respective datasets. This is our primary measure of quality, as it represents our approach's general ability to model text effectively.

**LAMBADA**   To explore our model's ability to extract useful information from context, we further evaluate the approach on the Lambada dataset (Paperno et al., 2016), a final-word prediction dataset where the correct answer is heavily dependent on detail revealed in context long before the final word. Given the entire context, we predict only the final word (i.e. space-delineated run of tokens) and compare against "gold"; a task is solved correctly if the final word exactly matches.

**HellaSwag**   To evaluate our model's knowledge and natural-language inference (NLI) capabilities, we perform evaluations against the HellaSwag dataset (Zellers et al., 2019)—a common-sense based NLI dataset. We concatenate each continuation against the premise and evaluate the perplexity of each. A task is solved correctly if the lowest-perplexity sequence is the target sequence.

**BLiMP**   To evaluate our model's syntax understanding, we evaluate our model's performance across all splits of the BLiMP dataset (Warstadt et al., 2020). The dataset contains pairs of lexically similar sequences, but only one of which is syntactically sound. A task is solved correctly if the model assigns lower perplexity to the grammatical sequence.

**AI2-ARC**   To evaluate our model's general in-context reasoning abilities, we measure our model's performance across both the easy and challenge slices of AI2-ARC (Clark et al., 2018). The dataset contains 4 choices of possible continuations of a given premise, only one of which is correct. A task is solved correctly if the model assigns lower perplexity to the correct continuation conditioned on the premise.

**PIQA**   Finally, to evaluate our model's knowledge and embodied common sense, we evaluate our model's performance on PIQA—an NLI style dataset for physical reasoning. (Bisk et al., 2020) As with HellaSwag, we concatenate each continuation

against the premise and evaluate the perplexity of each. A task is solved correctly if the lowest-perplexity sequence is the target sequence.

### H.1. Inference-Time Budget Scaling

Inference with short sequences on our method yields a distribution shift: if $\kappa$, the maximum block size, is kept the same for any block size, short sequences would be able to fork many more times than longer ones. This problem is especially prevalent during autoregression, where the initial input sequence is much shorter than the full block size.

To mitigate this, we scale the inference time forking budget $\kappa$, *proportionally* to the full-width block size. Specifically, we compute a training-time ratio $r = \frac{\kappa}{L}$ for training-time block size $L$ and maximum budget $\kappa$; at inference time, for an input of size $L'$, we set a temporary max budget $\kappa' = rL'$ for the top-k operation. This can also be understood as a "rolling top-k" operation that is iteratively updated at each token. This trick enables our method to work autoregressively with minimal performance degradation.

## I. Additional Methodological Comparisons

**COCONUT**   Hao et al. (2024) uses the last hidden state as the input embedding for serially generated chain of thought. Applying COCONUT requires a model trained with the ability to emit normal CoT before slowly removing the CoT traces to be continuous. Our model of adaptive computation is more generalizable since it does not require CoT labels and also more efficient as it does not require autoregressive serial decoding (our parallel thinking happens by copying existing residuals).

**Latent-SFT**   The Latent-SFT approach (Deng et al., 2025) requires a newly trained encoder to be trained in tandem with the decoder model as well as the introduction of explicit CoT labels which are gradually replaced by latent "compressed tokens." Our approach does not require CoT labels and can be applied directly at pretraining time, ensuring unification at train and test time.

**Pause Tokens**   Pause token training (Goyal et al., 2024) requires the insertion of pause tokens at oracle pause positions or by random insertions for pretraining. Our method implicitly learns where best to insert additional residuals, and correlates with the intuition that it inserts them at posterior uncertainty boundaries.

**Dot by Dot/Filler Tokens**   The dots approach (Pfau et al., 2024) requires explicit (verbalized) CoT supervision before the embeddings are slowly replaced by dots. The replacement rates and procedure is also designed through a carefully calibrated curriculum. Our approach can be directly applied without adaptation to pretraining and yields strong adaptive compute results.

**Looped Transformers / UTs**   Although the looped transformer approach (Saunshi et al., 2025; Dehghani et al., 2019) achieves adaptive computation, it cannot create more memory (i.e., residual channels) adaptively to store information and attend to during its model of computation.

**RLP**   This approach (Hatamizadeh et al., 2026) is a training-time optimization intervention and requires a two-stage pipeline and also a reference model to compare multiple rollouts. Our approach is an architecture experiment. Since our approach works directly with normal losses and pretraining, we believe our approach would be compatible with applying RLP on top.

