# OpenReview forum: "Thoughtbubbles: an Unsupervised Method for Parallel Thinking in Latent Space"
_ICML.cc/2026/Conference — ICML 2026 regular_

### Official Review · Reviewer_h1NV · 2026-03-12

**Soundness:** 4
**Presentation:** 3
**Significance:** 3
**Originality:** 3
**Overall Recommendation:** 5
**Confidence:** 4

**Summary:**

The paper introduces a method enabling a transformer to perform extra computation at certain tokens, by allowing the residual stream to fork. It can be seen as a more flexible extension of learning where to insert extra "thinking" tokens, as it allows capacity to be allocated more flexibly across the transformer stack (by making layer-specific decisions about forking or deleting streams, rather than making a single decision for the entire token/transformer stack once). The method is interesting but relatively complex (the process for computing the cumulative probabilities to make delete/fork decisions, modifications to attention, averaging across streams to make token predictions, etc.). The method is applied to pretraining and demonstrates improvements in zero-shot tasks and perplexity with respect to a standard transformer and wrt a version with a statically allocated residual stream (i.e., simply copying the stream multiple times).

**Compliance With Llm Reviewing Policy:**

Affirmed.

**Final Justification:**

Overall, I remain very positive about this work. It is an important contribution to the promising line of research on decoupling computation from tokens. I still feel the approach is slightly overcomplicated and there may be ways to simplify / streamline it, but I do not see this as a criticism so much as a question of what the next steps should be.

**Key Questions For Authors:**

1. See the question about post-training for reasoning tasks (or generally non-zero short experiments).

2. Intuitively, it seems that higher layers should be able to make more accurate decisions about whether extra “thinking” is needed. Could the higher layers for the previous token contribute to making the forking decision in lower layers for the next token? Something a bit like predicting a token that signals the need to think but in more 'soft' way? Or would this break parallelization in training and make the method too expensive?

**Limitations:**

yes

**Strengths And Weaknesses:**

1. I think the authors could provide better intuition re the connection between the use of “filler”/"thinking" tokens (eg as in Pfau et al paper) for thinking and the idea of forking residual streams. The connection eventually becomes clear, but the introduction itself does not explain it very effectivel:   there is a conceptual jump around line R042 .  However, generally the paper is well written.

2. While the idea of dynamically allocating extra thinking tokens seems nice, the approaches seems rather complicated (how the keep/fork probabilities computed; how attention is modified; etc). The discrete top-k decision can potentially create problems with gradient flow.

3. It may interesting to discuss relation to methods which use reasoning in pre-training (e.g., RLP https://arxiv.org/abs/2510.01265) and generally latent reasoning approaches (e.g., COCONUT, https://arxiv.org/abs/2412.06769).

4. This relation to pretraining-for-reasoning papers, which claim that such pretraining makes models better at post-training for reasoning problems, makes me wonder if something similar could be true here. Afterall, the objective these papers use is basically the same (reward there is log-prob of the next token). Have you considered fine-tuning it for reasoning tasks to see if it becomes stronger? The experiments in this submission seem to be only in the zero-shot setting (and convincing improvements are in perplexity, and not on all tasks).

I thought that the analysis where forks happen and the ablations (in appendix) were very nice.

---

> ### Author Rebuttal · Authors · 2026-03-27
>
> Thank you very much for your review, and noting the flexibility of our method.
>
> # Comparison to Related Work
> Thank you for the feedback. Below we explicitly call out the connection with dot-by-dot, specifically that while dot-by-dot uses a specialized curriculum to slowly replace dedicated chain-of-thought data, our method can be naively applied during pretraining. We will revise our introduction for camera-ready to use that language.
>
> Furthermore, soft tokens and COCONUT require the gradual elimination of explicitly trained CoT data while approach can be applied during pretraining, and RLP is a training optimization approach that is compatible with our approach which enables adaptive computation at the architecture level.
>
> **Dot by Dot/Filler Tokens** (Pfau et al., COLM 2024). The dots approach requires explicit (verbalized) CoT supervision before the embeddings are slowly replaced by dots. The replacement rates and procedure is also designed through a carefully calibrated curriculum. Our approach can be directly applied without adaptation to pretraining and yields strong adaptive compute results.
>
> **Pause Tokens** (Goyal et al., ICLR 2024). Pause token training requires the insertion of pause tokens at oracle pause positions or by random insertions for pretraining. Our method implicitly learns where best to insert additional residuals, and correlates with the intuition that it inserts them at posterior uncertainty boundaries.
>
> **COCONUT** (Hao et al., COLM 2025). COCONUT uses the last hidden state as the input embedding for serially generated chain of thought. Applying COCONUT requires a model trained with the ability to emit normal CoT before slowly removing the CoT traces to be continuous. Our model of adaptive computation is more generalizable since it does not require CoT labels and also more efficient as it does not require autoregressive serial decoding (our parallel thinking happens by copying existing residuals).
>
> **RLP** (Hatamizadeh et al., ICLR 2026). This approach is a training-time optimization intervention and requires a two-stage pipeline and also a reference model to compare multiple rollouts. Our approach is an architecture experiment. Since our approach works directly with normal losses and pretraining, we believe our approach would be compatible with applying RLP on top.
>
> # Complexity of the Approach / Gradient Flow Concerns
> We note that the complexity of our forking approach, barring a simple affine layerwise masking procedure, is isolated in a fixed amount of forking layers, and the actual top-k operations themselves do not take meaningful time based on a flamegraph measurement since they are done for a constant number of layers throughout the network. Our forking operations add negligible overhead compared to the extra residual channel for computation.
>
> For top-k induced gradient flow, we limit the extent of  this problem by limiting the number of forking layers to a finite amount, even when the number of layers increases.
>
> Upon acceptance we will release open source Jax and Torch codebases for reproducibility and extensions.
>
>
> # Downstream Fine-Tuning in Reasoning Tasks
>
> For downstream tasks, we performed mid-training in support of GSM8k with math heavy datasets and correspondingly found performance gains (Table 1). For camera-ready, we can conduct fine-tuning on various smaller-sized reasoning and problem-sorving datasets, such as MMLU and AI2-ARC.
>
> # Forking in later, rather than earlier, layers
>
> We will conduct this experiment and report the results accordingly; we believe forking starting only in later layers will result in less ability to do extra computation since there would be fewer layers to go before output.
>
> There is a balance to be struck in terms of finding where the residuals has been enriched enough to meaningfully fork (i.e., why we don’t fork immediately upon input) while there’s enough remaining computation to make the forks themselves meaningfully add sufficient additional computation.

---

> > ### Author Rebuttal · Reviewer_h1NV · 2026-04-02
> >
> > Thanks for the clarification. I’m keeping my recommendation to accept the submission.

---

### Official Review · Reviewer_6DHS · 2026-03-13

**Soundness:** 2
**Presentation:** 2
**Significance:** 2
**Originality:** 2
**Overall Recommendation:** 3
**Confidence:** 4

**Summary:**

The paper presents a method to conduct adaptive parallel computation in latent space during LLM reasoning. The model learns to fork or delete residual streams controlled by surviving scores that is learned by the model during pretraining. Experiments across model sizes from 150M to 1.9B showcase the effectiveness of the proposed method.

**Compliance With Llm Reviewing Policy:**

Affirmed.

**Final Justification:**

As justified in my rebuttal acknowledgment, I am not fully convinced by the novelty and training efficiency of the proposed method. I lean towards weak reject, but I acknowledge the contributions of the paper.

**Key Questions For Authors:**

Please refer to the points in the weaknesses section above.

**Strengths And Weaknesses:**

**Strengths:**

- The proposed method achieves adaptive allocation of the computation budget to emphasize on relatively high uncertainty regions that improves modeling efficiency.
- The method demonstrates stronger performance than the vanilla AR baseline on a set of downstream tasks.



**Weaknesses:**

- A careful analysis regarding the training and inference FLOPS and a fair comparison with baselines is missing. The only analysis in Appendix B only show the wall-clock time of the forward pass, and thus a rigorous analysis of the training FLOPS considering both the forward and backward is missing. Also, regarding the forward time, the proposed method takes about 1.4x running time than the AR baseline. Considering test-time scaling of LLMs [1], a fair comparison with baseline should be provided (i.e., performance comparison under the same computation or computation comparison with the same performance).
- The authors only compare with the vanilla AR baseline and the naive copied residual input baselines. More baselines should be compared or discussed, including reasoning with pause or thinking tokens [2], continuous space reasoning [3,4], looped transformer [5], etc.
- It is helpful to show how the model performance scales with the latent budget. It can be noticed that in Table 2, the performance of $\kappa=4L$ is sometimes worse than that of $\kappa=2L$. Could the authors explain the reason?
- The analysis of forks at regions of higher uncertainty is related to methods forcusing on inference time techniques, for example [6] and [7], while they have the advantage of very little training overhead. How does the proposed method compared with these inference-time methods?
- Minor points: some typos: line 198 cocatenated -> concatenated; Table 2 150M Ours ($\kappa= 2L$) ->  Ours ($\kappa= 4L$)



[1] s1: Simple test-time scaling

[2] Think Before You Speak: Training Language Models with Pause Tokens.

[3] Training Large Language Models to Reason in a Continuous Latent Space

[4] Soft Tokens, Hard Truth

[5] Hierarchical Reasoning Model

[6] Thought Calibration: Efficient and Confident Test-Time Scaling

[7] Deep Think with Confidence

---

> ### Author Rebuttal · Authors · 2026-03-27
>
> Thank you very much for your review! First, we'd like to emphasize that forking at regions of higher uncertainty is a consequence of our approach, not the overarching goal; it suggests that we are enabling the model to discover areas where additional computation is needed implicitly.
>
> # Detailed FLOPs analysis
> First, we note that our model of computation doesn’t require any assumptions regarding the post training quality or even chain-of-thought training at all, as does [1]. Our model is applied directly at pretraining time without specialized data, and learns implicitly.
>
> In Table 1 our method’s performance matches or exceeds $\kappa$ times the amount of training tokens in baseline (in particular, we test our model after 20B tokens of training against a baseline of 40B tokens), which is a FLOP, computation, and time upper-bound. Per suggestion, we will add the following FLOPs analysis to our article which shows that we use around double the FLOPs.
>
> In terms of the training FLOPs; let $d$ be our hidden dimension and $L$ be the block size:
>
> - attention projections costs $6Ld^{2}$
> - output projection costs $2Ld^{2}$
> - attention itself is quadratic in $4L^{2}d$
> - MLP for $16Ld^{2}$
>
> Forking projection, which maps inputs $L \times d$ to $d \times 2$, costs $4Ld$.
>
> Let the forking model's block size be $kL$. Thus the training performance ratio (baseline FLOPs / our FLOPs each block) would be:
>
> $$ \frac{24Ld^{2} + 4L^{2}d}{24kLd^{2} + 4k^{2}L^{2}d + 4kLd} = \frac{6d + L}{k\left(6d + kL + 1\right)} $$
> setting $d = 2048$, $L = 512$, $k = 2$, this would be around $0.481$, which as expected is around 2 times worth of FLOPs.
> In terms of inference time FLOPs, with an assumption of caching, each token would attend to two times the cached key-value pairs thus an analysis would follow similarly.
>
> # Contextualize Within Related Work
> We note that the model of adaptive computation we propose is different to most baselines in latent CoT literature. In particular,
>
> **COCONUT** (Hao et al., COLM 2025) & **Soft Tokens** (Butt, et al. 2025). COCONUT uses the last hidden state as the input embedding for serially generated chain of thought. Applying COCONUT requires a model trained with the ability to emit normal CoT before slowly removing the CoT traces to be continuous. Our method requires no CoT labels and avoids needing serial decoding through parallel forking.
>
> **Pause Tokens** (Goyal et al., ICLR 2024). Pause token training requires the insertion of pause tokens at oracle pause positions or by random insertions for pretraining. Our method implicitly learns where best to insert additional residuals, and correlates with the intuition that it inserts them at posterior uncertainty boundaries.
>
> **Looped Transformers / UTs** (Saushi et al., ICLR 2025). Although the looped transformer approach achieves adaptive computation, it cannot create more memory (i.e., residual channels) adaptively to store information and attend to during its model of computation. Our architecture-driven approach is complementary, and we do hope to integrate with (MoE)UT in future work, which will give us both recurrence and parallel computation.
>
> **HRM** (Wang et al., 2025). This approach is an encoder-only model with a two-block hierarchical decoding implemented using encoder blocks; in requires special problems (e.g., sudoku) that is amenable to this encodre-only structure. Our approach is an autoregressive transformer that requires no specialized data to train, and in our approach adaptive computation time emerges without supervision, and in particular without expensive additional training of a Q-function.
>
> **Deep Think with Confidence** (Fu et al., ICLR 2026) & **Thought Calibration** (Wu et al., EMNLP 2025). These are a post-training methods for supervising existing chain-of-thought models to control the amount of thinking, but in order to do any reinforcement-learning style post-training, explicit CoT must first emerge which does not arise sufficiently only from unsupervised pretraining. Our method implicitly induces adaptive computation with only pretraining.
>
> # Relationship of $\kappa$ and performance
> We note that the difference in performance between these two budgets, especially at a small scale such as 300M parameters, can be much noisier than our scaled-up experiments. We expect the relationship between the scaling ratio and performance to increase monotonically, but with diminishing returns when additional residual channels become less useful.
>
> # Inference-Time Methods
> We want the model to be able to learn the reasoning traces without hand-designed heuristics. During pretraining, we do not have an accessible chain of thought nor steerable prompting behavior, yet both of these are necessary for inference-time scaling methods.
>
> We note that, even though our method presents a 2x training time overhead, we show that even with half the training budget it outperforms the baseline, rendering it basically "free" in a resource constrained setting.

---

> > ### Author Rebuttal · Reviewer_6DHS · 2026-04-05
> >
> > Thank you for your rebuttal, which basically addresses my major concerns. I will adjust my score to 3. My remaining concerns are internal novelty and training overheads of the method - which is not possible to address during rebuttal.

---

> > > ### Author Response · Authors · 2026-04-05
> > >
> > > Thank you very much for your response! With regards to your remaining concerns:
> > >
> > > > internal novelty
> > >
> > > In addition the novelty of our approach in terms of its paradigm and formulation, we present here a few sources of architectural novelty of our implementation.
> > >
> > > **Forking**. Instead of adding new residual streams from the beginning of computation or skipping layers, we dynamically allocate token streams in the middle of the forward pass: cloning and pruning streams as needed to maintain a fixed budget while allowing adaptive computation.
> > >
> > > **Attention Attenuation Used for Training**. Although previous methods have used attention attenuation as a regularization tool, we leverage the novel insight that learnable attenuators are also a signal for what the model finds strongly important. This allows us to enable dynamic forking without auxiliary losses or additional supervision.
> > >
> > > > training overheads
> > >
> > > As we stated in the rebuttal, we note that, as seen in Table 1 of our results, even though our method incurs a 2x FLOPs and wall clock training time overhead, it performs competitively against a baseline approach when using only half of the training data.
> > >
> > > This implies our model is basically "free" in a resource constrained setting in terms of training overhead, as a FLOPs-matched variant (half the tokens, double the compute) is competitive to the baseline.
> > > In contrast to the baseline, our model is additionally helpful in a data-constrained setting, as training with the full token budget (and thus 2x FLOPs) results in the significantly better performance we observed.

---

### Official Review · Reviewer_Vazc · 2026-03-23

**Soundness:** 3
**Presentation:** 2
**Significance:** 4
**Originality:** 4
**Overall Recommendation:** 5
**Confidence:** 4

**Summary:**

This paper explores yet another promising avenue for scaling through the adaptive computation paradigm which is loosely analogous to chain-of-thought in latent space. In contrast to previous works, such as Coconut (https://arxiv.org/abs/2412.06769) that reuse output hidden representations as new latent embeddings for implicit reasoning, the Thoughtbubbles method selectively inserts or removes latent-thinking token representations across a subset of hidden layers inside the stack of Transformer blocks. Therefore, each latent representation can last just for a few consecutive layers. This redirects the pre-allocated reasoning budget to tokens of the sequence where it’s needed the most. The authors empirically prove that this allocation method outperforms both compute-matched vanilla Transformer and latent-reasoning models with fixed uniform compute budget allocation.

**Compliance With Llm Reviewing Policy:**

Affirmed.

**Key Questions For Authors:**

1. Line 160, right column: what is the argument of $\nu_{\theta}$ function?

2. Do I understand correctly that a fork of the token can produce a fork of itself, and so on, forming the eponymous bubble of tokens?

3. Can a token have $p_{fork} >p_{keep}$ which would potentially lead to retaining a fork of the token while eliminating the original?

4. Do I understand correctly that the number of forks of an original token grows maximally by one at each additional layer?

5. Lines 166-169: why “the rightmost token does not have forced-maximum score of 1” if $p_{keep}$ for such tokens is always 1?

**Limitations:**

Yes.

**Strengths And Weaknesses:**

**Strengths**
* The main strength of this research in my opinion is its uniqueness and originality of the approach. I believe it markedly differs from the existing solutions that I’m aware of as it’s more flexible and adaptive due to automatic dynamic allocation of the reasoning budget.

* I feel excited about this approach because it should be both easily implementable and transferable for real LLMs, and it offers a simple and intuitive path (fork/keep scores and top-K selection) for dynamic allocation and management of latent thinking budget.

* I believe this implicit reasoning paradigm can be combined with explicit chain-of-thought reasoning and improve the performance of next generations of frontier open-weights reasoning models. I can also foresee that this work can inspire future follow-up research.

* The empirical analysis is comprehensive. It’s especially appealing that the authors compare the Thoughtbubbles variant trained both on 20B and 40B tokens to a vanilla Transformer trained on 40B tokens. It demonstrates that  their method clearly wins even if we equate the latent-reasoning tokens (Thoughtbubbles)  with explicit additional pre-train tokens (Transformer).

* The authors provide detailed configurations of the models and experiments which is beneficial for reproducibility and future research.

**Weaknesses**

From my points of view, the only weaknesses of this paper lie in the area of presentation/ exposition, and they can be easily remediated:


**1.**

I would classify this work as related to the latent chain-of-thought subfield. This is a burgeoning but already rich in literature subfield (judging by https://arxiv.org/abs/2505.16782, especially Figure 2). I missed the extended related work section (perhaps in the Appendix) which would list previous and concurrent advantages and position your architecture among the most relevant alternatives. For example, I would like to read an explicit account of architectural differences between your approach and Coconut (https://arxiv.org/abs/2412.06769) or Latent-Chain (https://arxiv.org/abs/2510.15522) among other works.

**2.**

Neither a theoretical nor empirical motivation is provided for the attenuation process described in Section 2.4. Why is it required and does it actually help boosting performance additionally to just adding the adaptive computation via forked tokens?

For example, I can somewhat intuitively derive that $P^{(k)}$ inside the softmax in formula 8 can redistribute attention weights back to the original tokens. But an explicit motivation, as well as explanation of why the specific form (logarithm of $P^{(k)}$) was chosen for this term, would be helpful.

Also, why is gating by $P^{(k)}$ required for V and MLP? It shouldn't influence the fork/keep/discard decisions and just uniformly downgrades the magnitude of select token representations in comparison with others.

It would benefit the work if you could provide either theoretical motivation/ derivation why attenuation is required in all of these places or conduct ablation studies which demonstrate the boost in performance after attenuation.

**3.**

Notation/ typos:

* It was difficult to understand what the notion “residual stream” exactly means in the interpretation of this paper. Usually, in deep learning literature it means $X_l$ in $X_{l+1}=X_l + f(X_l)$, i.e. inputs of the layer including all of the tokens. In this work, the definition is a representation of a single token or its copies. Please consider giving a formal definition of this notion, as well as forking, at the start of the paper.

* Line 73, left column: I’m confused whether your intention here was to really mention attenuation in the sense of Section 2.4, or you really meant the fork/keep/discard procedure from Section 2.3.

* Equation 1: I couldn’t understand at first that RHS is the output of the sigmoid. It’d be better to swap LHS and RHS of the equation.

* Line 191: What did you mean by “residual whites”?

* Equations 8 and 9 combined make an impression that gating by  $P^{(k)}$ gets applied twice to V. Is it a typo?

* Line 323: *confers

* Line 685: is -> does

**4.**

A minor concern not related to exposition. One of the strongest validation results in Table 2 exhibits that even 2L variant of ThoughtBubbles almost uniformly surpasses even Copy-5 fixed-duplication Transformer variants across scales and benchmarks. However, the pre-training was conducted on just 2.5B tokens, and I can attest from personal experience that this training duration might be insufficient to compare the modeling performance of architectural modifications. A modification with best results at 2.5B tokens can fall behind significantly at longer training durations. Table 1, which lists results for 20-40B tokens training, doesn’t have the data for fixed duplication factor variants (Copy-X).

Therefore, I would like to ask whether you could additionally train and publish the results for these Copy-X variants in Table 1, to definitively demonstrate which approach is better after prolonged train duration. I understand that you might not have access to compute resources or they might be limited at the moment, so I’d like to state that fulfilling this request is not required but would be nonetheless appreciated (if you have the capacity).

---

> ### Author Rebuttal · Authors · 2026-03-27
>
> Thank you very much for your review! We are particularly excited for you noting the originality of our work, and the robustness of our analysis.
>
> # Contextualize Within Related Work
> Agreed that there should be related work discussion; we will add the following to appendix, some comments on how other approaches differ from ours:
>
> **COCONUT** (Hao et al., COLM 2025). COCONUT uses the last hidden state as the input embedding for serially generated chain of thought. Applying COCONUT requires a model trained with the ability to emit normal CoT before slowly removing the CoT traces to be continuous. Our model of adaptive computation is more generalizable since it does not require CoT labels and also more efficient as it does not require autoregressive serial decoding (our parallel thinking happens by copying existing residuals).
>
> **Latent-SFT** (Deng et al., 2025, arxiv.org/abs/2510.15522). The Latent-SFT approach requires a newly trained encoder to be trained in tandem with the decoder model as well as the introduction of explicit CoT labels which are gradually replaced by latent "compressed tokens.” Our approach does not require CoT labels and can be applied directly at pretraining time, ensuring unification at train and test time.
>
> # The Attention Attenuation Mechanism
> In Appendix C., we report an ablation that removes the masking on the attention weights and find that MLP masking alone is insufficient for the performance gains we observed. On the other hand, removing MLP masking will enable tokens to write to residual streams that have very low cumulative scores, which may not be present in a future layer; simply masking attention score computation does not prevent the model from rendering a big update to a particular residual channel anyways, and we found in preliminary experiments that our approach with masking both works better than no masking at all, where the scores vanish to be small such that training becomes unstable.
>
> Consider the term for attention scores: $$ \mathrm{softmax}\left(\frac{Q^{(k)} {K^{(k)}}^{T} + \mathbf{1} \log\left(P^{(k)}\right)^{T}}{\sqrt{d_{\mathrm{model}}}}\right) $$
> Now think about key-query pairs $(i,j)$. Let $s_{ij} = q_{i}^{(k)T} k_{j}^{(k)} / \sqrt{d_{\mathrm{model}}}$ for brevity. Then: $$ \frac{\exp\left(s_{ij} + \frac{\log P^{(k)}{j}}{\sqrt{d_{\mathrm{model}}}}\right)}{\sum{j'} \exp\left(s_{ij'} + \frac{\log P^{(k)}{j'}}{\sqrt{d_{\mathrm{model}}}}\right)} $$
> Through exponents: $$ \frac{\exp\left(\frac{\log P^{(k)}{j}}{\sqrt{d_{\mathrm{model}}}}\right) \cdot \exp\left(s{ij}\right)}{\sum_{j'} \exp\left(\frac{\log P^{(k)}{j'}}{\sqrt{d_{\mathrm{model}}}}\right) \cdot \exp\left(s{ij'}\right)} $$
> which gives: $$ \frac{(P^{(k)}{j})^{1/\sqrt{d_{\mathrm{model}}}} \exp\left(s{ij}\right)}{\sum_{j'} (P^{(k)}{j'})^{1/\sqrt{d_{\mathrm{model}}}} \exp\left(s{ij'}\right)} $$
> $(P^{(k)}{j})^{1/\sqrt{d{\mathrm{model}}}}$ thus is exactly a multiplicative scale on the attention matrix indexed to keys. Intuitively, this operation softly removes the tokens with low score from the attention.
>
> # Gating by Pk gets applied twice to V
> It is indeed applied twice. Notice that the first mask is indexed to values and the second mask is indexed to queries: if either the residual being attended to or the residual being updated is “almost missing” via a small $P_k$, the attention update will be small.
>
> # Copy-n at scale
> Thank you, and we appreciate you noting our performance gains here. We will work to run this approach for camera-ready, but the training process itself at sufficient scale is compute availability bound which may not be resolved during the review period.
>
> # "Residual Streams"
> After forking we treat the forked tokens and non-forks indistinguishably, and thus a single “residual stream” refers to the values $x_i^{(j)}$  used in each layer.
>
> # Notation
> > Line 73 meaning of attentunation
>
> We mean in both counts, we believe (and early results demonstrate that) attention attenuation nor MLP attenuation alone is insufficient to enable the magnitude of gains we see here
>
> >  Equation 1: swap LHS and RHS of the equation.
>
> Thank you for the suggestion. We will include this for camera-ready.
>
> > “residual whites”
>
> Typo, our apologies. “Residual writes”, that is, updates to the residual channel.
>
> >  Line 323: *confers,  Line 685: is -> does
>
> Noted, thank you.
>
> # Questions
>
> > What's $\nu_\theta$'s argument
>
> Its a fixed embedding, learned per layer, rather than a function.
>
> > Do I understand correctly that a fork of the token can produce a fork of itself, and so on, forming the eponymous bubble of tokens?
>
> Yes!
>
> > Can a token have $p_{fork} > p_{keep}$
>
> Yes! Other than the rightmost token, for whom $p_{keep}$ is $+\infty$.
>
> > Number of forks grow maximially by one
>
> No, since forks can fork.
>
> > rightmost token doesn't have forced-max
>
> To clarify, this means that the cumulative score used for attenuation, etc., isn't the one forced to one and is instead the value it had before we forced it to be 1 for topk.

---

> > ### Author Rebuttal · Reviewer_Vazc · 2026-04-03
> >
> > My feedback has been addressed. I still have a couple of remaining comments:
> >
> > W1: Please include more related works in the new section, not just the two you mentioned.
> >
> > W2:  “from rendering a big update to a particular residual channel anyways” – do you mean during forward pass or backward pass? Please discuss explicitly what are the drawbacks of a big update to the residual stream with low cumulative score. I understand that if a token with low “keep” score has large magnitude, it should not influence the next layers in the model, because this token would be pruned via top-K selection.
> >
> > I remain positive about this work and vote for its acceptance. Dear authors, please include your answers and discussions from the rebuttal in the final version.

---

> > > ### Author Response · Authors · 2026-04-03
> > >
> > > Thank you very much! Re your comments:
> > >
> > > > W1: Please include more related works in the new section, not just the two you mentioned.
> > >
> > > Agreed, here's a full list we plan to add.
> > >
> > > **COCONUT** (Hao et al., COLM 2025) & **Soft Tokens** (Butt, et al. 2025). COCONUT uses the last hidden state as the input embedding for serially generated chain of thought. Applying COCONUT requires a model trained with the ability to emit normal CoT before slowly removing the CoT traces to be continuous. Our method requires no CoT labels and avoids needing serial decoding through parallel forking.
> > >
> > > **Pause Tokens** (Goyal et al., ICLR 2024). Pause token training requires the insertion of pause tokens at oracle pause positions or by random insertions for pretraining. Our method implicitly learns where best to insert additional residuals, and correlates with the intuition that it inserts them at posterior uncertainty boundaries.
> > >
> > > **Dot by Dot/Filler Tokens** (Pfau et al., COLM 2024). The dots approach requires explicit (verbalized) CoT supervision before the embeddings are slowly replaced by dots. The replacement rates and procedure is also designed through a carefully calibrated curriculum. Our approach can be directly applied without adaptation to pretraining and yields strong adaptive compute results.
> > >
> > > **Looped Transformers / UTs** (Saushi et al., ICLR 2025). Although the looped transformer approach achieves adaptive computation, it cannot create more memory (i.e., residual channels) adaptively to store information and attend to during its model of computation. Our architecture-driven approach is complementary, and we do hope to integrate with (MoE)UT in future work, which will give us both recurrence and parallel computation.
> > >
> > > **HRM** (Wang et al., 2025). This approach is an encoder-only model with a two-block hierarchical decoding implemented using encoder blocks; in requires special problems (e.g., sudoku) that is amenable to this encodre-only structure. Our approach is an autoregressive transformer that requires no specialized data to train, and in our approach adaptive computation time emerges without supervision, and in particular without expensive additional training of a Q-function.
> > >
> > > **Deep Think with Confidence** (Fu et al., ICLR 2026) & **Thought Calibration** (Wu et al., EMNLP 2025). These are a post-training methods for supervising existing chain-of-thought models to control the amount of thinking, but in order to do any reinforcement-learning style post-training, explicit CoT must first emerge which does not arise sufficiently only from unsupervised pretraining. Our method implicitly induces adaptive computation with only pretraining.
> > >
> > > **Latent-SFT** (Deng et al., 2025, arxiv.org/abs/2510.15522). The Latent-SFT approach requires a newly trained encoder to be trained in tandem with the decoder model as well as the introduction of explicit CoT labels which are gradually replaced by latent "compressed tokens.” Our approach does not require CoT labels and can be applied directly at pretraining time, ensuring unification at train and test time.
> > >
> > > > do you mean during forward pass or backward pass
> > >
> > > Forward pass. The insight is that, as you noted, not adding additional masking means models may be spending large updates on residual channels with low cumulative scores. This amounts to doing work on residuals that will soon be disappeared (and in particular not attended to much during each layer.) This is an inefficient allocation of additional computation, and may even result in gradient flow problems since this large update doesn't have a correspondingly large gradient signal (due to future cumulative scores being smaller).

---

### Decision · Program_Chairs · 2026-04-30

**Decision:**

Accept (regular)

**Comment:**

This paper proposes a new architecture for allocating variable amounts of compute to different tokens in Transformers. Unlike existing  early-exit-style strategies which apply a different number of computations to each token (whose early stopping criterion is typically predicted by the token), this paper proposes to expand the computation "horizontally" by forking the residual stream.

The reviewers generally agreed that this is an interesting and novel idea, and backed up by solid experiments that demonstrate the promise of the approach against comparable baselines. Several reviewers suggested that the authors could have better contextualized the work compared to works such as COCONUT and Pause tokens. During the rebuttal, the authors have adequately responded to this point in my opinion.

My biggest worry is that these models are relatively undertrained, e.g., 700M param models trained on 2.5B tokens, which is fewer tokens than even Chinchilla-optimal. It is unclear whether these gains would persist at more modern regimes, where LLMs are (over)trained far beyond Chinchilla optimal. However, given the novelty of the architecture, I think that there is enough empirical experiments here for this to be a worthwhile contribution to the community.